# The role of palbociclib on the alterations in CDKN2, CCNE1, E2F3, MDM2 expressions as target genes of miR-141

Mohammad Ali Baziyar[1], Arshad Hosseini[1]*, Farinush Jandel[2]

**1** Department of Medical Biotechnology, School of Allied Medicine, Iran University of Medical Sciences, Tehran, Iran, **2** Blood Transfusion Research Center, High Institute for Research and Education in Transfusion Medicine, Tehran, Iran

* hoseini.a@iums.ac.ir

**Data Availability Statement:** ll relevant data are within the manuscript and its Supporting Information files.

**Funding:** The author(s) received no specific funding for this work.

## Abstract

### Introduction

According to WHO, Breast cancer is widely considered to be the first or second cause of cancer-related death almost universally. Cell cycle disruption, either in the form of uncontrolled expression of cyclins or because of the suspension in negative regulatory proteins (CDK inhibitors), was found to cause breast cancer. Palbociclib as specific CDK4/6 inhibitor is used for the treatment of ER$^+$ metastatic cancers. In this study, we are looking to investigate the effect of palbociclib on breast cancer cells and evaluate the changes in the expression of some genes involved in the cell cycle as target genes of *miR-141* after treatment with this drug. We used MCF7 as functional estrogen and non-invasive and MDA-MB-231 cell lines as triple-negative type of breast cancer and a model for more aggressive.

### Method & materials

MCF7 and MDA-MB-231 cell lines were *cultured* in DMEM medium. After counting cells and measuring viability, Palbociclib was administered at varying doses using the IC50 obtained from MTT, with the treatment given at two time points of 24 and 72 hours. RNA was extracted from untreated and treated cells and RNAs were converted to cDNA in the end. Gene expression changes were investigated by real-time PCR. Data management and analysis were conducted using GraphPad Prism 5.01 software.

### Result and conclusion

Among investigated genes, *E2F3* gene was not significantly affected by Palbociclib in any of cell lines and time points. Besides, the expression of *CCNE1* gene was significantly suppressed. It seems this drug was unable to reduce the expression of *MDM2* gene significantly in triple negative (MDA-MB-231) cancer cells; however, a decrease was observed in luminal A (MCF-7) cells. *CDKN2A* and *miR-141* genes expression increased significantly after treatment which can be aligned with palbociclib in proliferation inhibition.

**Competing interests:** The authors have declared that no competing interests exist.

## 1. Introduction

According to the World Health Organization (WHO), breast cancer is one of the leading causes of cancer-related deaths worldwide. It affects 2.3 million people each year, making it the most common cancer among adults. Breast cancer is categorized into four subtypes based on the presence of estrogen receptor (ER), progesterone receptor (PR), and HER-2: luminal A, luminal B, HER-2 positive, and triple-negative [1]. The most prevalent subtype is luminal A, which is characterized by being ER-positive (ER$^+$), PR-positive (PR$^+$), and HER-2 negative. Luminal A breast cancer generally has a good prognosis. While endocrine therapy is effective for treating luminal A breast cancer [2], chemotherapy is the only effective treatment for triple-negative breast cancer, which is known for its aggressive nature and high metastasis rate [3, 4]. cell lines such as MCF-7 and MDA-MB-231 are commonly used to represent these cancer subtypes in research studies [5].

Cell cycle disruption, either through uncontrolled expression of cyclins or the suspension in negative regulatory proteins(CDK inhibitors), was found to cause breast cancer [6, 7]. Retinoblastoma phosphorylation by the CDK4/6 cyclin complex leads to its dissociation from the E2F transcription factor, promoting cell cycle progression [8]. However, this process may be downregulated by p21 and p27 known as CDK inhibitors from the CIP-KIP family, or by P16 member of the INK family [9]. Moreover, by inhibiting the activity of *P53* [10], *MDM2* can be downregulated by P14, which is encoded by *CDKN2A* [11]. In this context, a class of noncoding RNAs, named microRNAs play an important role in cancer biology by cell cycle regulation, proliferation, Metastasis and apoptosis [12, 13]. For example, the miR-141/200a cluster has been found to reduce retinoblastoma protein phosphorylation, leading to a decline in the G1/S transition phase. Meanwhile, overexpression of this microRNA further promotes CDK inhibitor proliferation and G1 arrest [14]. Therefore, downregulation of *miR-141* in breast cancer is significant [15]. The specified genes in this study including *CCNE1*, *CDKN2A*, *E2F3* and *MDM2*, are targets of this microRNA based on databases such as Mirbase and Mirwalk.

Palbociclib, Ribociclib and Abemaciclib, as specific CDK4/6 inhibitors, obtained FDA approval for the treatment of ER-positive metastatic cancers [16], which combined with endocrine agents like fulvestrant and letrozole, may improve survival rates [17]. Evidence suggests that palbociclib, in the presence of functional retinoblastoma protein binds to the ATP-binding pocket of CDK4/6 preventing the function of CDK4/6/cyclin D Complex [18]. As a consequence of this action, Rb phosphorylation is stopped and cell cycle progression from G1 to S is blocked. following palbociclib treatment, a decrease in P107, accumulation of cyclin D1 & E1 and decrease in E2F-regulated protein occur [19].

From a clinical perspective, in addition to conventional methods of breast cancer treatment such as chemotherapy drugs and hormone therapy, immunotherapy drugs can also have a promising future [20, 21]. Nevertheless, treatment results are often hindered by drug resistance and lack of reliable biomarkers. Therefore, a better understanding of these potential mechanisms might have gone some way toward dealing with breast cancer [22].

In this study, we performed experiments on MDA-MB-231 and MCF-7 cell lines to evaluate the effects of Palbociclib on cell growth and to analyze the changes in the expression of genes involved in the cell cycle and *miR-141*.

## 2. Method & materials

### 2.1 Cell lines and reagents

The MDA-MB-231 and MCF-7 cell lines used in this study were obtained from the Iranian Biological Resource Center. The cells were cultured in Dulbecco's Modified Eagle Medium/

F12 (DMEM, BioIdea) with 4.5 g/L glucose L-glutamine, supplemented with 10% fetal bovine serum (FBS, BioIdea), 100 U/mL penicillin (BioIdea, Iran) and 0.1 mg/mL streptomycin (BioIdea, Iran). The cells were maintained in a humidified incubator at 37°C with 5% $CO_2$. Once the cells reached appropriate density, they were washed with Phosphate-buffered saline (PBS) before subculturing with phenol-free trypsin/ethylene diamine tetra acetic acid (EDTA 0.25%). Palbociclib, which was purchased from MedChem Express (MCE, USA), was dissolved in 1ml sterile dimethyl sulfoxide (DMSO, DNA biotech, Iran) and stored at -20°C.

## 2.2 MTT cell viability assay

Breast cancer cells were seeded at a concentration of $1\times10^4$ cells/mL in 100 μL per well in a 96-well plate and were incubated overnight at 37°C. The study aimed to determine the half-maximal inhibitory concentration (IC50) of palbociclib. After 24 hours, cells were treated with gradual doses of the drug and analyzed at 24, 48 and 72 hours. Then, 100 μL of MTT was added to each well and incubated for 4 hours. The fluorescence absorbance was measured at a wavelength of 530/590 nm using a DANA microplate ELIZA reader (Iran).

## 2.3 RNA extraction and qRT-PCR

To determine the impact of drugs on the expression of target genes, $3 \times 10^5$ cells were seeded in 6-well culture plates. The next day, the cell lines were treated with palbociclib at concentrations corresponding to the IC50 obtained from the viability test. The concentrations were 18 μM in MDA-MB-231 cells and 49 μM in MCF-7 cells. Treatment was performed at two time points, 24 and 72 hours. After treatment, cells were collected from each plate in a fresh tube and stored at -80 °. Experiments for each control and treated group were conducted in triplicate.

According to the manufacturer's instructions, total RNA was extracted using Trizol reagent (Yektatajhiz, Iran). The RNA quality and purity of each sample were validated using a Nano-Drop spectrophotometer at a 260/280 nm ratio (NanoDrop Technologies, Thermo Fisher Scientific, Inc.). The RNA was then converted into cDNA using 2 μl of random hexamer primers in a microtube containing 2 μg of RNA at a total volume of 11.4 μl using Notarkib kit (Iran). Due to the short sequences of miRNAs, cDNA synthesis of *miR-141* was carried out using stem-loop specific reverse transcriptase primers. The expression level of *miR-141* was quantified using specified primers followed by qRT-PCR analysis. Similarly, quantitative Real-time PCR was also performed for *E2F3*, *CDKN2A*, *CCNE1*, and *MDM2* genes in both transfected and untreated cells using the SYBR Green master mix kit (Qiagen, Iran). Primers were designed using the primer blast site on the NCBI website and AllelID software. Also, in order to check secondary structures such as double-stranded and hairpin structures, primer sequences were checked with AllelID and Primer3. Primers were purchased from Metabion (Germany). Table 1 presents an overview of the forward (F) and reverse (R) primer sequences of *miR-141* (NR_029682.1), *E2F3* (NM_001949.5), *CDKN2A* (NM_000077.5), *CCNE1* (NM_001238.4), and *MDM2* (NM_002392.6) used in real-time PCR. The *U6* and *GAPDH* mRNA genes were considered as the internal standard control gene by specific primers for *miR-141* and target genes, respectively. Analysis for gene expression patterns of treated and untreated cells was performed with the $2^{-\Delta\Delta CT}$ method.

## 2.4 Bioinformatics

MicroRNA target prediction: Different microRNA databases including MirBase, miRWalk, and miRanda, were investigated to determine whether the *CCNE1*, *E2F3*, *CDKN2A*, and *MDM2* as CDK4/6/cyclin dependent are the direct targets of *miR-141*. GeneMANIA is a fast

**Table 1. Forward (F) and reverse (R) primer sequences of miR-141 and targeted genes used in real-time PCR.**

| Oligonucleotide | Sequence | PCR product size |
|---|---|---|
| *miR-141* | | |
| Forward primer | 5' GCGCGTAACACTGTCTGGTA' 3 | 95(bp) |
| Reverse primer | 5' CGTGGTTAGGGTCCGAGGTA' 3 | |
| *E2f3* | | |
| Forward primer | 5' GCCTGACTCAATAGAGAGCCTAC' 3 | 155(bp) |
| Reverse primer | 5' AGTCTTTGGAAGCGGGTTTAGG' 3 | |
| *CDKN2A* | | |
| Forward primer | 5' GAGTCAACGGATTTGGTCGT' 3 | 155(bp) |
| Reverse primer | 5' GACAAGCTTCCCGTTCTCAG' 3 | |
| *CCNE1* | | |
| Forward primer | 5' TGCCACCTGCGTGAAGAAG' 3 | 160(bp) |
| Reverse primer | 5' ACCTATTCCGTTACACACTTTGC' 3 | |
| *MDM2* | | |
| Forward primer | 5' ACTTCGGCATCAGTGGACAG' 3 | 131(bp) |
| Reverse primer | 5' GACATCAGAGCGGACATCATATC' 3 | |

gene network construction and function prediction tool. The results were subsequently used to predict the interactions between *TP53* and gene targets of *miR-141*.

## 2.5 Statistical analysis

GraphPad Prism 9.0.0 software was used for data management and analysis. For determining the significance of differences, two-way ANOVA analysis of variance followed by Tukey's comparison test was performed, and p-values less than 0.05 were considered statistically significant. P-values of less than 0.05, less than 0.01, and less than 0.001 were marked by *, ** and ***, respectively.

## 3. Results

### 3.1 Predicted target genes of miR-141

Based on searches of online databases such as TargetScan, miRWalk, miRDB, and GeneMANIA, we selected *MDM2*, *CDKN2A*, *CCNE1*, and *E2F3* as genes involved in the cell cycle and targets of *miR-141* (Fig 1).

### 3.2 Palbociclib inhibited cell viability and proliferation of breast cancer cells

The results shown in (Fig 2) indicate that the cell viability of both MCF-7 and MDA-MD-231 cell lines decreased significantly after treatment with Palbociclib in a dose and time-dependent manner compared to the untreated control cells. This data shows that palbociclib further inhibits cell proliferation by decreasing the IC50 value over different exposure periods (24–72 hours).

### 3.3 Gene expression profile in palbociclib-treated breast cancer cell lines

In order to gain a detailed understanding of the effect of palbociclib on the CDK4/6 signaling pathway, expression levels of E2F3, CCNE1, CDKN2A, MDM2 genes, and miR141 at 24 and

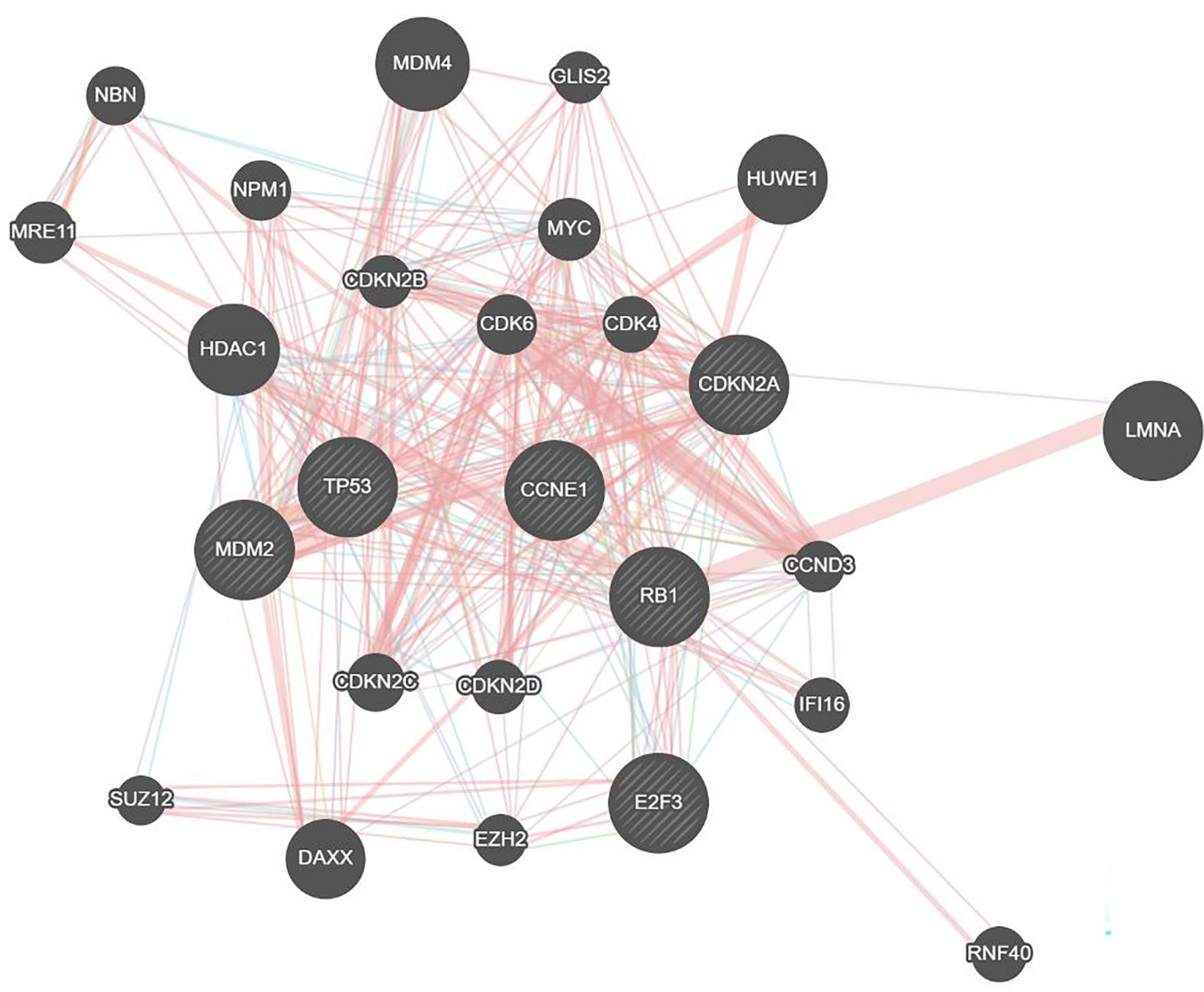

**Fig 1. Target prediction and gene interaction.** Target genes of miR-141 that are involved in cell cycle.

72 hours were measured after treatment on MCF-7 and MDA-MB-231 cell lines. These cell lines represent ER$^+$, HER2$^+$, and triple-negative breast cancer cells, respectively.

The expression of genes at mRNA levels was investigated via qRT-PCR. Relative quantitation of genes was normalized to *GAPDH* (and *U6* against *miR-141*) as a housekeeping gene.

As shown in the bar chart (Fig 3), the expression levels of *CDKN2A* and *miR-141* in both cell lines and time points were significantly increased. In contrast, *CCNE1* and *MDM2* gene expression tended to decrease after palbociclib treatment, except for *MDM2* gene, which was not significantly affected in MDA-MB-231 cell line. The *MDM2* gene in MCF-7 cell line seems to have a greater decrease in gene expression at 24h compared to the other time point. Also, the analysis did not reveal any significant differences in *E2F3* gene between two cell lines and time points of 24 and 72 hours after treatment.

As observed in (Fig 3E), palbociclib induced a significant increase in the expression of *miR-141* in the MDA-MB-231 cell line at 24h (1.6 fold) more than at 72h (0.5 fold). Consequently, this drug had a greater increasing effect on the MDA-MB-231 cell line.

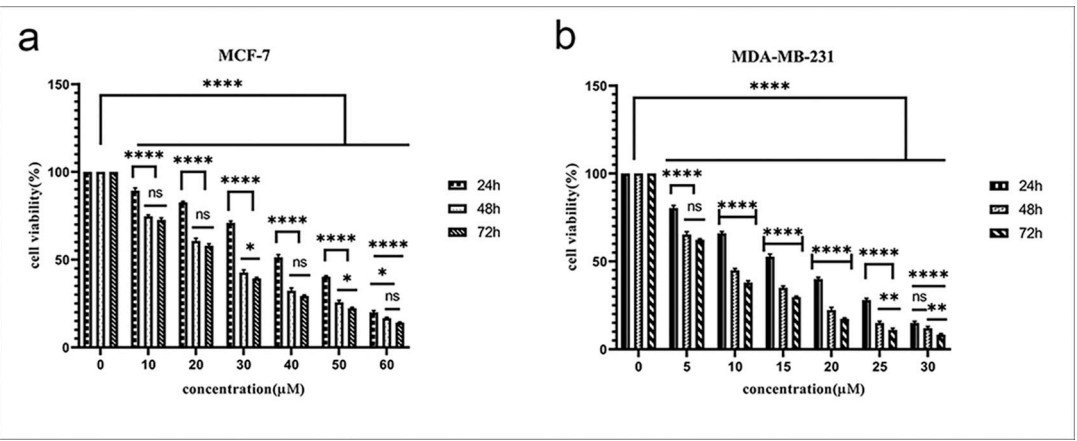

**Fig 2.** Effect of palbociclib on cell viability in (a) MCF-7 and (b) MDA-MD-231 cell lines. By analyzing the results of the MTT test, we reached a concentration of 18 μM in MDA-MB-231 cells and 49 μM in MCF-7 cells as IC50.

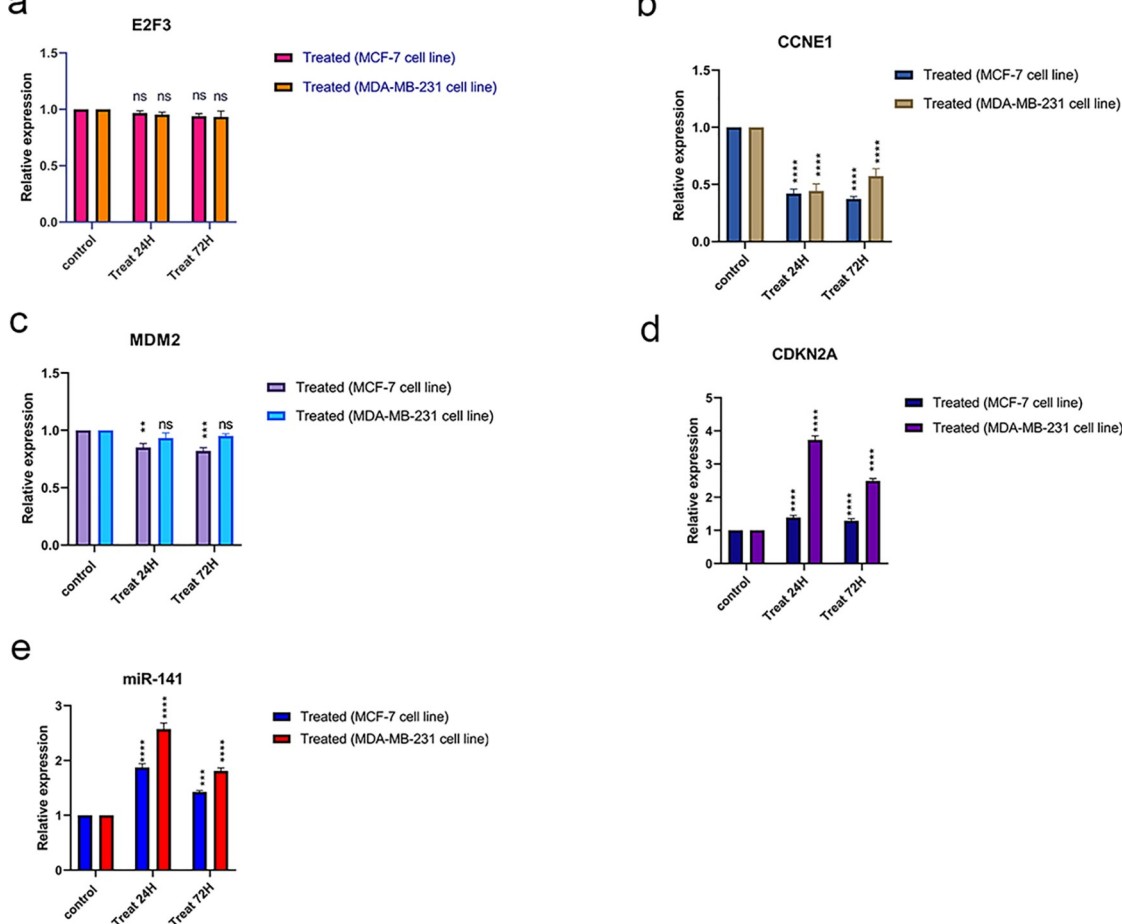

**Fig 3.** The relative mRNA levels of (a) E2F3, (b) *CCNE1*, (c) *MDM2*, (d) *CDKN2A*, (e) *miR-141* in MDA-MB-231 and MCF-7 cell lines. mRNA levels after treatment by palbociclib were analyzed by qRT-PCR. For 24 and 72 h. cells were exposed to 18 and 49$\mu M$ of palbociclib, respectively. Data are signified as the mean ± SD. The significance of the difference between the palbociclib-treated vs. untreated MDA-MB-231 and MCF-7 cells was determined using an unpaired T-Test. The difference between control and treated cells was considered significant at *p = 0.05, **p = 0.01, and ***p = 0.001.

The *CDKN2A* gene in the MDA-MB 231 cell line showed the most increased expression among genes which was 1.35 fold at 72h and 2.61 fold at 24h compared to control. In addition, the *miR-141* gene had a higher expression level in the MCF-7 cell line. The changes in the *CCNE1* gene expression in each time point and cell lines equally showed a significant decrease of approximately 0.4 fold compared to the control.

## 4. Discussion

This research has identified dose and time-dependent decreases in cell viability in both cell lines. The qRT-PCR results showed a significant increase in the expression of *miR-141*, a single-strand RNA that regulates gene expression at the post-transcriptional level, after palbociclib treatment in both cancer types. Among the target genes of this miRNA which was investigated in this study, *E2F3*, *CCNE1* and *MDM2* showed decreased expression due to palbociclib. However, the *E2F3* gene was not significantly affected by drug in any of the cell lines and time points. Reducing the expression of these genes leads to cell cycle arrest and prevents cancer cell growth. Conversely, the expression of the *CCNE1* gene as the cell cycle promoter was significantly suppressed. Palbociclib did not significantly reduce the expression of the *MDM2* gene in triple negative (MDA-MB-231) cancer cells; however, a decrease was observed in luminal A (MCF-7) cells. As expected, *CDKN2A* gene expression as a tumor suppressor gene in both types of cancer increased significantly after treatment which can be aligned with palbociclib in proliferation inhibition.

These results support previous research which found that in MDA-MB-231 cells treated with palbociclib, genes that contributed to cell cycle upregulation and persistence were significantly suppressed, Instead, genes promoting cell cycle arrest showed increased expression [23]. In a study conducted by Ping Li. et al, it has been identified that compared to surrounding intact tissues, *miR-141* is downregulated in breast tumor tissues, besides overexpression of *miR-141* can disrupt cell proliferation, migration and invasion [15]. As mentioned above, *miR-141* expression increased after treatment. Collectively, these results suggest that the rise in *miR-141* may have contributed to the increase inhibition of cell growth in line with palbociclib.

In the cell cycle cascade, CDK4/6-cyclin D complex has been confirmed as an E2F activator by phosphorylating Rb, enabling the synthesis of S phase initiator proteins and conducts positive feedback for E2F gene expression proliferation [24]. Out of E2F family, *E2F3*, along with *E2F1* and *E2F2*, typically binds to pRb and controlled through it. While *E2F4* and *E2F5* are mostly controlled through P107 and P130. Phosphorylation of these regulatory proteins leads to the activation of their target genes [25]. The effectiveness of CDK4/6 inhibitors on Rb phosphorylation has been demonstrated in a report by Finn et al. [26]. Also, a direct effect on reducing Rb expression was noticed in the study of Choupani et al. [27]. Michaloglou et al. pointed out that Palbociclib can inhibit the function of E2F by cooperating with mTORC1/2 inhibitor (Vistusertib), which has been shown to alter protein levels in the MCF-7 cell line [28]. It is also stated that the *E2F3* expression at the protein and mRNA level was significantly downregulated in gastric cancer cells after *miR-141* transfection [29]. As regards the qRT-PCR results, the expression level of *E2F3* was not significantly affected after treatment with palbociclib. It may have provided insights into the theory that the observed nonsignificant result may differ at the protein level.

Contrary to ER-negative cell lines such as MDA-MB-231, all ER-positive cell lines such as MCF-7 express higher levels of *MDM2* mRNA [30]. Based on real-time PCR results on *MDM2* gene expression, consistent with the report mentioned, palbociclib seems to considerably reduce MDM2 expression in ER+ luminal A cell line. Regarding the inhibitory effect of *CDKN2A* on *MDM2* through the ARF protein (p14arf) in the cell cycle. according to the

increased expression of *CDKN2A* after treatment, it can be assumed that *CDKN2A* works aligned with palbociclib to inhibit *MDM2* at the protein level.

Due to the role of *CCNE1* in cell cycle progression, boosting the *CCNE1* expression in cancer cells is predictable. Turner et al. used the PALOMA-3 study to assess palbociclib's efficacy by *CCNE1* mRNA expression, which confirmed the prognostic role of *CCNE1* in metastatic cancer. In this way, the lower expression of *CCNE1* has been certified to be associated with the progression of the palbociclib potency [31]. This study provides evidence of a significant reduction of *CCNE1* expression in both cancer cell lines. As well as drug impression on mRNA level, according to cyclin E/CDK2 settlement at the *MDM2* downstream and inhibitory role of *MDM2* on P53 [32, 33], it can be concluded that palbociclib reduces *MDM2* expression, which causes P53 enhancement and organized inhibitory effect on Cyclin E/CDK2 complex at the protein level.

P16 (*CDKN2A*) protein, an INK4 family member, is a tumor suppressor that plays a significant role in CDK4/6–cyclin D restriction, contributing to G1 arrest through direct binding to CDK4 [34]. Along with this, *CDKN2A* and *Rb1* are tumor suppressors that affect the manner of reduction of *MDM2* and the proliferation of P53 [35]. Baghdadi et al. 2019 failed to ascertain the positive effect of palbociclib on cell cycle inhibition in patients suffering from advanced pancreatic and biliary cancers with *CDKN2A* loss or mutation [36]. As mentioned earlier, this study has found a significant increase in *CDKN2A* expression in two cell lines. Through the P16 protein, *CDKN2A* can regulate Rb phosphorylation following CDK4/6 attachment, which can force the cell to stay in the G1 phase and lead to cell cycle arrest. This may have been in line with the palbociclib goal.

The analysis of palbociclib on the expression of signaling pathway genes undertaken here has extended our knowledge to new ways to control and treat breast cancer. However, the study was limited by the lack of information on protein level through western blotting which due to the insufficiencies, a more extensive study is suggested.

## 5. Conclusions

Based on our results, palbociclib inhibits cell viability in a dose and time-dependent manner in both MDA-MB-231 and MCF-7 cell lines. Besides, Palbociclib affects the expression of some genes that contributed to cell cycle cascade at the mRNA level and also *miR-141* expression as regulator mRNAs in these cell lines. Overexpression of *miR-141* results in an interruption in cell proliferation, migration, and invasion.

This study indicates that Palbociclib stops the cell cycle in the G1-S phase by reducing the expression of the *CCNE1* gene. The reduction of *MDM2* gene expression after treatment with Palbociclib in MCF-7 cell line will lead to cell growth inhibition through *P53*. Also, by increasing the expression of *CDKN2A* gene through P16 protein, it forces the cell to stay in G1 phase.

## Supporting information

**S1 Dataset. Raw data used for graph (related to Fig 1).**
(XLSX)

**S2 Dataset. Raw data used for data analysis (related to Fig 2).**
(XLSX)

## Acknowledgments

The authors gratefully acknowledge the Cell and Molecular Research Center at the Faculty of Allied Medicine, Iran University of Medical Sciences, for providing help, service, and devices.

We thank Davood Jafari (Department of Medical Biotechnology, School of Allied Medicine, Iran University of Medical Sciences, Tehran, Iran) for experimental technical help.

## Author Contributions

**Conceptualization:** Mohammad Ali Baziyar.

**Data curation:** Arshad Hosseini.

**Formal analysis:** Mohammad Ali Baziyar, Arshad Hosseini.

**Investigation:** Arshad Hosseini.

**Methodology:** Mohammad Ali Baziyar.

**Project administration:** Mohammad Ali Baziyar.

**Resources:** Mohammad Ali Baziyar.

**Software:** Mohammad Ali Baziyar, Farinush Jandel.

**Supervision:** Arshad Hosseini.

**Validation:** Arshad Hosseini.

**Writing – original draft:** Mohammad Ali Baziyar, Farinush Jandel.

**Writing – review & editing:** Mohammad Ali Baziyar, Farinush Jandel.

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
