## [Decision Letter · Decision Letter 0]

19 Jan 2024

PONE-D-23-42371The role of Palbociclib on the alterations in CDKN2, CCNE1, E2F3, MDM2 expressions as target genes of miR-141PLOS ONE

Dear Dr.Arshad Hosseini,

Thank you for submitting your manuscript to PLOS ONE. After careful consideration, we feel that it has merit but does not fully meet PLOS ONE’s publication criteria as it currently stands. Therefore, we invite you to submit a revised version of the manuscript that addresses the points raised during the review process.

We look forward to receiving your revised manuscript.

Kind regards,

Sudhir Kumar Rai, Ph.D

Academic Editor

PLOS ONE

Journal Requirements:

2. We note that your Data Availability Statement is currently as follows: [ll relevant data are within the manuscript and its Supporting Information files.]

Reviewers' comments:

Reviewer's Responses to Questions

**Comments to the Author**

1. Is the manuscript technically sound, and do the data support the conclusions?

Reviewer #1: Partly

Reviewer #2: Partly

2. Has the statistical analysis been performed appropriately and rigorously? 

Reviewer #1: Yes

Reviewer #2: Yes

3. Have the authors made all data underlying the findings in their manuscript fully available?

Reviewer #1: Yes

Reviewer #2: No

4. Is the manuscript presented in an intelligible fashion and written in standard English?

Reviewer #1: No

Reviewer #2: Yes

5. Review Comments to the Author

Reviewer #1: 1. Authors are suggested to re-write the manuscript in grammatically correct and in standard academic English with non-ambiguous sentence structure. The majority of the sentences are confusing and hard to understand what authors exactly want to convey.

2. In "Results and Conclusion" section in the abstract, please write in a non-ambiguous, coherent, and clear manner. Especially regarding MDM2 gene in lines 16 and 17.

3. In the last paragraph of "Introduction" section, after line 57, please provide brief sentences on "what you are going to do in this paper" before you move into the methods section.

4. Authors are suggested to conduct an additional protein detection experiments, such as Western Blot to support their assumptions that are made on "Discussion section".

5. In "Discussion" section in lines 136 and 137, please clarify using an additional sentence why Palbociclib reducing the expression of these genes are important?

6. From line 136 to 143, the remark regarding CDKN2A is clear, however, the previous conclusion with MDM2 in lines 139 and 140 is not clear. You data shows that the expression of E2F3, CCNE1 and MDM2 were reduced upon Palbociclib treatment, however, only the expression of CCNE1 was significantly reduced, thus, indicating the cell cycle arrest. Please conclude clearly about MDM2.

7. In line 165, if you can not see significant difference in mRNA level, it is unlikely that the difference can be significant at protein level. Looking for the proteins could be more confirmatory, thus authors are suggested to add protein identification experiments, such as western blot.

8. In "Conclusion" section from the line 196 onwards, include all the findings in brief. In line 197, Clarify "affect" means "increase" or "decrease". Clarify "Some genes"; please mention the name of the genes. Clarify "contributed to cell cycle cascade"; clarify which gene expression affects what stage of cell cycle, mentioning if it "arrests" or "promotes" the cell cycle.

Reviewer #2: Title: The role of Palbociclib on the alterations in CDKN2, CCNE1, E2F3, MDM2 expressions as target genes of miR-141

Major revision:

The results for Section 2.4. Bioinformatics, are missing. The authors have mentioned that they have attempted microRNA target prediction, but no results have been provided.

Figures 2 and 3 show the same results. The authors can use one of the two figures to show the results and avoid confusion.

A few minor comments:

Line 62: Correct ‘CO2’. 2 should be subscript.

Line 65,76,76: Correct the temperature notation to ‘°C’.

Figure 2: There is no pie chart. Please correct.

Section4. Correct ‘discussion’ to ‘Discussion’

Use uniform notation for Figures throughout the manuscript.

6. PLOS authors have the option to publish the peer review history of their article (what does this mean?). If published, this will include your full peer review and any attached files.

Reviewer #1: No

Reviewer #2: No

---

## [Author Response · Author response to Decision Letter 0]

18 Apr 2024

Response to Reviewer #1

1. Authors are suggested to re-write the manuscript in grammatically correct and in standard academic English with non-ambiguous sentence structure. The majority of the sentences are confusing and hard to understand what authors exactly want to convey.

Response: Thank the Reviewer for careful consideration, The language of the manuscript was reviewed, and Some changes were made.

2- In "Results and Conclusion" section in the abstract, please write in a non-ambiguous, coherent, and clear manner. Especially regarding MDM2 gene in lines 16 and 17.

Response: In this study, we investigated the changes in gene expression in two cell lines, MCF-7 and MDA-MB-231, after treatment with palbociclib. Regarding MDM2, there was a decrease in expression in the MCF-7 cell line, but no significant expression change occurred in the MDMA-MB-231 cell line.

3. In the last paragraph of "Introduction" section, after line 57, please provide brief sentences on "what you are going to do in this paper" before you move into the methods section.

Response: Thank the Reviewer for careful consideration, the correction was made.

4. Authors are suggested to conduct an additional protein detection experiments, such as Western Blot to support their assumptions that are made on "Discussion section".

Response: Thanks for your very insightful and detailed suggestion. We agree, but this study project is related to the master's thesis and due to financial and time shortages related to university regulations, it was not possible for us to carry out this proposed project.

5. In "Discussion" section in lines 136 and 137, please clarify using an additional sentence why Palbociclib reducing the expression of these genes are important?

Response: Thank you very much for your detailed and useful comments. Reducing the expression of these genes leads to cell cycle arrest and prevents the growth of cancer cells.

6. From line 136 to 143, the remark regarding CDKN2A is clear, however, the previous conclusion with MDM2 in lines 139 and 140 is not clear. You data shows that the expression of E2F3, CCNE1 and MDM2 were reduced upon Palbociclib treatment, however, only the expression of CCNE1 was significantly reduced, thus, indicating the cell cycle arrest. Please conclude clearly about MDM2.

Response: Thank the Reviewer for careful consideration, We have described this in Discussion section (line 167-169).

7. In line 165, if you can not see significant difference in mRNA level, it is unlikely that the difference can be significant at protein level. Looking for the proteins could be more confirmatory, thus authors are suggested to add protein identification experiments, such as western blot.

Response: Thank you for your detailed and useful comments. According to the decrease in MDM2 gene expression and increase in CDKN2A gene expression after palbociclib treatment. By inhibiting the cyclin D-CDK4/6 D complex, Rb proteins are not phosphorylated and they are not separated from E2F transcription factors, and E2F transcription factors cannot continue their activity. Also, by increasing miR-141 gene expression, it inhibits the expression of its target genes (including E2F3) at the post-transcriptional level.

8. In "Conclusion" section from the line 196 onwards, include all the findings in brief. In line 197, Clarify "affect" means "increase" or "decrease". Clarify "Some genes"; please mention the name of the genes. Clarify "contributed to cell cycle cascade"; clarify which gene expression affects what stage of cell cycle, mentioning if it "arrests" or "promotes" the cell cycle.

Response: We thank Reviewer 1 for this valuable suggestion and we agree. We modified our Discussion sections (lines 227–231)

Response to Reviewer #2

Major comments: The results for Section 2.4. Bioinformatics, are missing. The authors have mentioned that they have attempted microRNA target prediction, but no results have been provided.

Figures 2 and 3 show the same results. The authors can use one of the two figures to show the results and avoid confusion.

Response: In accordance with reviewer 2’s suggestion, We removed Figure 3 and added Figure 1 as results for Section 2.4. Bioinformatics.

minor comments: 

Line 62: Correct ‘CO2’. 2 should be subscript.

Line 65,76,76: Correct the temperature notation to ‘°C’.

Figure 2: There is no pie chart. Please correct.

Section4. Correct ‘discussion’ to ‘Discussion’

Use uniform notation for Figures throughout the manuscript.

Response: Thank Reviewer 2 for careful consideration, the correction was made.

We hope that the revisions in the manuscript and our accompanying responses will be sufficient to make our manuscript suitable for publication in PLOS ONE.

We shall look forward to hearing from you at your earliest convenience.

---

## [Decision Letter · Decision Letter 1]

29 Apr 2024

PONE-D-23-42371R1The role of Palbociclib on the alterations in CDKN2, CCNE1, E2F3, MDM2 expressions as target genes of miR-141PLOS ONE

Dear Dr. Hosseini,

Thank you for submitting your manuscript to PLOS ONE. After careful consideration, we feel that it has merit but does not fully meet PLOS ONE’s publication criteria as it currently stands. Therefore, we invite you to submit a revised version of the manuscript that addresses the points raised during the review process.

We look forward to receiving your revised manuscript.

Kind regards,

Sudhir Kumar Rai, Ph.D

Academic Editor

PLOS ONE

Journal Requirements:

Reviewers' comments:

Reviewer's Responses to Questions

**Comments to the Author**

1. If the authors have adequately addressed your comments raised in a previous round of review and you feel that this manuscript is now acceptable for publication, you may indicate that here to bypass the “Comments to the Author” section, enter your conflict of interest statement in the “Confidential to Editor” section, and submit your "Accept" recommendation.

Reviewer #1: (No Response)

Reviewer #2: All comments have been addressed

2. Is the manuscript technically sound, and do the data support the conclusions?

Reviewer #1: Yes

Reviewer #2: Yes

3. Has the statistical analysis been performed appropriately and rigorously? 

Reviewer #1: Yes

Reviewer #2: Yes

4. Have the authors made all data underlying the findings in their manuscript fully available?

Reviewer #1: Yes

Reviewer #2: Yes

5. Is the manuscript presented in an intelligible fashion and written in standard English?

Reviewer #1: No

Reviewer #2: Yes

6. Review Comments to the Author

Reviewer #1: 1. Regarding the previous comment, “In the Results and Conclusion section in the abstract, please write in a non-ambiguous, coherent, and clear manner. Especially regarding MDM2 gene in lines 16 and 17”, I understand that palbociclib treatment reduced the expression of MDM2 and CCNE1 expression in MCF-7 cells. However, the first sentence in the results and conclusion section in your abstract is still ambiguous. Please correct the first sentence and avoid repetition in the latter sentences in your manuscript.

2. Academic standard English is still lacking and ambiguity is still present, please revise the manuscript thoroughly.

Reviewer #2: The authors have diligently addressed the points I raised. The revision has significantly enhanced the manuscript. I recommend that it be accepted for publication.

7. PLOS authors have the option to publish the peer review history of their article (what does this mean?). If published, this will include your full peer review and any attached files.

Reviewer #1: No

Reviewer #2: No

---

## [Author Response · Author response to Decision Letter 1]

13 Jun 2024

Please find attached a revised version of our manuscript “The role of Palbociclib on the alterations in CDKN2, CCNE1, E2F3, MDM2 expressions as target genes of miR-141”, which we would like to resubmit for publication as a Research Article in PLOS One.

Your comments and those of the reviewers were highly insightful and enabled us to greatly improve the quality of our manuscript. In the following pages are our point-by-point responses to each of the comments of the reviewers as well as your own comments. Revisions in the text are shown using yellow highlight in the marked-up copy of the manuscript.

Response: Following your suggestion, We removed reference 3 and checked the reference list to ensure that it is complete and correct.

Response to Reviewer #1

1.Regarding the previous comment, “In the Results and Conclusion section in the abstract, please write in a non-ambiguous, coherent, and clear manner. Especially regarding MDM2 gene in lines 16 and 17”, I understand that palbociclib treatment reduced the expression of MDM2 and CCNE1 expression in MCF-7 cells. However, the first sentence in the results and conclusion section in your abstract is still ambiguous. Please correct the first sentence and avoid repetition in the latter sentences in your manuscript. 

Response: Thank the Reviewer for careful consideration, the correction was made.

2. Academic standard English is still lacking and ambiguity is still present, please revise the manuscript thoroughly. 

Response: Thank the Reviewer for careful consideration, The manuscript was revised and changes were made.

We hope that the revisions in the manuscript and our accompanying responses will be sufficient to make our manuscript suitable for publication in PLOS ONE.

---

## [Editor Report · Decision Letter 2]

20 Jun 2024

The role of Palbociclib on the alterations in CDKN2, CCNE1, E2F3, MDM2 expressions as target genes of miR-141

PONE-D-23-42371R2

Dear Dr.Arshad Hosseini,

We’re pleased to inform you that your manuscript has been judged scientifically suitable for publication and will be formally accepted for publication once it meets all outstanding technical requirements.

Kind regards,

Sudhir Kumar Rai, Ph.D

Academic Editor

PLOS ONE

---

## [Editor Report · Acceptance letter]

1 Jul 2024

PONE-D-23-42371R2 

PLOS ONE

Dear Dr. Hosseini, 

I'm pleased to inform you that your manuscript has been deemed suitable for publication in PLOS ONE. Congratulations! Your manuscript is now being handed over to our production team.

Kind regards, 

on behalf of

Dr. Sudhir Kumar Rai 

Academic Editor

PLOS ONE